# Barbed Dental Ti6Al4V Alloy Screw: Design and Bench Testing

**DOI:** 10.3390/ma16062228

**Published:** 2023-03-10

**Authors:** Keila Lovera-Prado, Vicente Vanaclocha, Carlos M. Atienza, Amparo Vanaclocha, Pablo Jordá-Gómez, Nieves Saiz-Sapena, Leyre Vanaclocha

**Affiliations:** 1CIRU-IMPLANT, S.L., Avenida Cornellà, 2-BJ, Esplugues de Llobregat, 08950 Barcelona, Spain; 2Department of Surgery, School of Medicine, University of Valencia, 46010 Valencia, Spain; 3Biomechanical Engineer, Biomechanics Institute of Valencia, Polytechnic University of Valencia, 46022 Valencia, Spain; 4Hospital General Universitario de Castellón, 12004 Castellón de la Plana, Spain; 5Hospital General Universitario de Valencia, 46014 Valencia, Spain; 6Medius Klinik, Ostfildern-Ruit Klinik für Urologie, Hedelfinger Strasse 166, 73760 Ostfildern, Germany

**Keywords:** bone–implant interactions, osseointegration, titanium implants, porous implants, 3D printing

## Abstract

Background context. Dental implants are designed to replace a missing tooth. Implant stability is vital to achieving osseointegration and successful implantation. Although there are many implants available on the market, there is room for improvement. Purpose. We describe a new dental implant with improved primary stability features. Study design. Lab bench test studies. Methods. We evaluated the new implant using static and flexion–compression fatigue tests with compression loads, 35 Ncm tightening torque, displacement control, 0.01 mm/s actuator movement speed, and 9–10 Hz load application frequency, obtaining a cyclic load diagram. We applied variable cyclic loadings of predetermined amplitude and recorded the number of cycles until failure. The test ended with implant failure (breakage or permanent deformation) or reaching five million cycles for each load. Results. Mean stiffness was 1151.13 ± 133.62 SD N/mm, mean elastic limit force 463.94 ± 75.03 SD N, and displacement 0.52 ± 0.04 SD mm, at failure force 663.21 ± 54.23 SD N and displacement 1.56 ± 0.18 SD mm, fatigue load limit 132.6 ± 10.4 N, and maximum bending moment 729.3 ± 69.43 mm/N. Conclusions. The implant fatigue limit is satisfactory for incisor and canine teeth and between the values for premolars and molars for healthy patients. The system exceeds five million cycles when subjected to a 132.60 N load, ensuring long-lasting life against loads below the fatigue limit.

## 1. Introduction

Tooth loss, and bone degradation associated with traumas and age-related degeneration, are important public health problems [1]. Bone density decreases after age 30, resulting in a reduction of up to 40% in mechanical resistance [2]. Moreover, osteoporosis is common in older adults [3] and likely contributes to tooth loss.

Dental implants are designed to replace the root of a missing tooth [4] and serve to hold the new artificial tooth in its correct position [5]. These implants are a safe and durable solution for the loss of one or more teeth. Researchers have developed countless dental implants and insertion techniques to meet a wide variety of patients’ medical conditions [6,7,8]. Unfortunately, the presence of osteoporotic bone reduces success with dental implants due to loosening [9].

Odontologists have developed implants based on titanium [10], tantalum [7], and zirconia [11], among others. These bioinert materials show contact osteogenesis, so the implant’s union with the bone is mainly mechanical, producing so-called “biological fixation” [12]. As a result, the bone grows orderly in direct contact with the implant but does not fix to it [13]. Titanium-based implants are advantageous because of osseointegration, namely a structural and function connection between implant and bone. Titanium’s other advantages include relatively easy manufacturing and high mechanical resistance. Not surprisingly, most dental implants are made from titanium [8,14]. Other important properties are its biocompatibility and high corrosion resistance [15].

Implant surface roughness and porosity foster osseointegration, particularly needed in the case of osteoporotic edentulous patients [16]. Surface porosity allows bone tissue growth within the implant, integrating it with the bone, which makes it possible to ensure secondary stability [10,17]. Porous structures are also highly osteoconductive, with fast vascularized fibrous tissue invasion and substantial accelerated (by three to five times) incorporation into surrounding bone compared to other rough surfaces [18]. The pores act in two ways [19]: mechanical coupling with nearby bone and, on the other hand, directly and indirectly influencing the cells’ metabolism [17]. Pre-osteoblasts show a kind of “poro-philia”, i.e., increased gene expression, differentiation, and mineralization related to signals received from the implant’s surface [20].

Implants with complex porous structures are most easily manufactured using 3D-printing technology.

Primary stability is a third crucial dental implant feature, which is essential for optimal long-term osseointegration [6]. Once said stability takes place, its quality is important, as well as how fast and biomechanically strong it can be, depending in most cases on the presence of suitable anchoring systems [21,22]. The implant itself must be strong to avoid breakage.

Considering the above discussion, we developed a new dental implant whose primary anchoring system is based on clockwise threading. Two outgoing legs deployed by a screw provide secondary stability. Additionally, this dental implant has a porous structure to enhance osseointegration. The mechanical goals of the implant are (1) to provide an immediate anchor to the cortical bone once inserted and (2) a porous surface that allows bone on-growth and in-growth. This design should enable good primary stability in older patients with osteoporosis and type IV bone [23] as well as adequate long-term osseointegration.

Once we designed the dental implant and decided on its material, titanium, manufacturing it was the next challenge. Additive 3D printing from a 3D CAD file is a good alternative when manufacturing an object with complex geometry and a relatively small size, such as our dental implant [24,25].

To evaluate the mechanical characteristics of our dental implant, we performed static and flexion–compression fatigue tests.

## 2. Material and Methods

The device was manufactured out of a grade five titanium alloy (Ti6Al4V, elasticity module 1100 MPa) and has three components: the dental implant body, the barrette, and the deployment screw. The dental implant body consists of (1) an implant–abutment connection thread in the implant’s head, (2) two anchoring system exit holes, (3) a double thread with a 1 mm pitch to facilitate implant insertion, (4) a self-drilling tip thread, (5) a porous zone to improve osseointegration, (6) a 16° angle hexagonal implant–abutment connection with Morse taper fixation to guarantee optimal connection sealing, and (7) an internal thread for anchor insertion and attachment connection (Figure 1). The two implant body exit holes are opposite each other, corresponding to the arrangement of the deployable wings of the anchorage.

The implant body has a porous, trabecular-shaped structure in the middle area of the external thread.

The external threaded area is a double 1 mm thread to reduce bone removal during implant insertion, reduce heat generation, and improve torque insertion in low-density bone.

The implant body self-drilling tip allows changing the implant’s orientation during insertion, thus enabling correct parallelism between implants, and optimizing their placement.

The implant–abutment connection area has an angle of 16° to reduce the friction between the walls.

The implant–abutment connection uses a Morse taper fixation to reduce the micro gap between the implant and the abutment connection, minimizing bacterial microleakage, decreasing bone tissue reabsorption, and stabilizing soft tissues with a better long-term aesthetic result. In addition, this connection type absorbs the vibration and pressure exerted on the abutment, minimizing the potential for prosthetic screw loosening.

The upper hole connects the anchor itself and the insertion screw.

The deployment screw has a hexagonal head connection, a shank, and a thread. The thread allows the screw to move in and out inside the implant (Figure 1, number 7), pushing the barrette externally. The shank connects the screw with the barrette, and the hexagonal connection facilitates screw manipulation, dental implant insertion, and removal using the corresponding instruments (Figure 2).

The barrette is inserted into the implant body. When the deployment screw is threaded in, the barrette’s legs bend outwards and protrude through the implant’s holes mentioned above (Figure 3). These two legs support against the mandible or maxilla cortical walls, and a clip at its center connects it with the screw.

With the assembly in position, as shown in Figure 4D, the deployment screw is threaded into the implant body’s internal channel. Then, the barrette’s legs bend and expand outside the implant, holding against the lateral and medial mandibular and maxilla cortical bone (Figure 4E). The barrette’s legs inside the dental implant body can be recovered by inverting the process and unscrewing the deployment screw.

The barrette’s legs are 1.5 mm in diameter, have a blunt tip on both sides, and extend 4 mm from the implant once deployed (Figure 5).

We can insert the abutment or cover screw once the deployment screw is fully inserted and secured, and the barrette’s legs are deployed. This cover screw holds the artificial tooth in place.

The implant is patented in Spain, the USA (Patent number 16/956594), and Brazil.

The implant barrette and the dental implant body were additively manufactured to achieve the middle zone porosity of the latter (Figure 6). Next, we machined the implant body’s internal thread as well as the deploying screw and gave it an Allen hexagonal head.

Figure 7 shows the set implanted to replace an incisor tooth in a human jaw reconstructed using a 3D simulation program.

### Implant Testing

We evaluated our implant through static and flexion–compression fatigue tests, following the UNE-ENISO 14801:2017 standard “Dentistry. Implants. Dynamic fatigue test for endosseous dental implants” [26]. All testing was performed using an INSTRON 8874/135 universal testing machine (Norwood, MA, USA) at 22 °C and 58% humidity.

For the test, we used three components: the dental implant, a universal conical base connection RP 3 mm REF 38217 (Nobel Biocare, Zürich, Switzerland), and a securing screw (Figure 8A–C, respectively).

We inserted the dental implant into a polymethyl methacrylate resin with bone-like rigidity. We placed this construct inside the stainless-steel machine clamp between the testing machine actuators, applying compression loads as shown in Figure 9. The tightening torque applied to the implant connection screw was 35 Ncm.

We performed the test by displacement control, 0.01 mm/s actuator movement speed, and the test end condition was implant failure.

We defined the bending moment as M = sin30 × 11 × F, M = bending moment and F = force.

We applied cyclical loads through a haversine (rectified sine) with the amplitude, preload, and maximum load values indicated in Table 1.

The load application frequency was 9–10 Hz. First, we constructed a cyclic load diagram (S-N Wohler curves) [27] to evaluate the assembly’s flexion–compression fatigue resistance. Then, we tested the implants at variable cyclic loading of predetermined amplitude and recorded the number of loading cycles until failure. The test was performed by flexion–compression force control, applying cyclical loads using a haversine (rectified sine) with amplitude, preload, and maximum load. The end of the test was either the implants’ components’ failures (breakage or permanent deformation) or reaching 5 million cycles for each load value.

We tested thirty-five dental implants.

## 3. Statistical Analysis

We used R software ((R Development Core Team 2018) R: A language and environment for statistical computing. R Foundation for Statistical Computing, Vienna, Austria, URL https://www.R-project.org, accessed on 1 January 2022) in combination with the Deducer user interface library (I. Fellows, “Deducer: A Data Analysis GUI for R”, Journal of Statistical Software, 2012; 49(8): 1–15, doi:10.18637/jss.v049.i08) [28]. We calculated the mean and standard deviation. We considered them statistically significant values if *p* < 0.005.

## 4. Results

In the geometric arrangement indicated in the ISO 14801:2017 standard, mean (SD) values for stiffness were 1151.13 ± 133.62 N/mm, at an elastic limit force of 463.94 ± 75.03 N and displacement of 0.52 ± 0.04 mm. The mean failure force was 663.21 ± 54.23 N, and dental implant displacement failure occurred at a mean of 1.56 ± 0.18 SD mm.

Implant failures were either permanent connection system deformations or dental implant ruptures (Figure 10).

We show below the charge cycle diagram. Our dental implants (screw plus barrette deployed) had an elastic force limit of 463.94 ± 75.03 SD N and a failure force of 663.21 ± 54.23 N.

Figure 11 shows the S-N Wholer curve results. The fatigue limit (maximum loads at which failure does not occur for infinite loading cycles of trial limit) was 132.6 ± 10.4 N.

In the geometric arrangement indicated in the ISO 14801:2017 standard, the maximum bending moment was 729.3 ± 69.43 mm/N. Table 2 shows the number of cycles the dental implant supported for each load.

All failures during cyclic loading occurred at the actuator–implant interface, not the implant itself.

## 5. Discussion

We designed a dental implant prototype optimized for use in patients with grade IV osteoporotic bone, which is prevalent in the elderly population [29]. The discussion below highlights key design features that may be advantageous. We are unaware of any other dental implant that rests in the mandible or maxilla lateral cortical bone [14], improving the primary fixation capacities.

Titanium, the material selected for our implant, is used in 91% of dental implants [14,15]. While zirconia might have better osseointegration [8,11] it is less resistant to permanent failures (cracking) [30]. Similarly, while porous tantalum has greater biocompatibility, lower corrosion, and greater porosity than titanium [7], resulting in a greater osseointegration capacity [7,31], manufacturing dental implants with tantalum is technically cumbersome [32]. Only one tantalum implant is available in the EU market, and this implant is a combination of tantalum and titanium [7]. In addition, we have found no data on the mechanical resistance of other porous dental implants to compare with ours.

Dental implant design dramatically impacts primary stability and long-term osseointegration [6,33], therefore reinforcing fixation elements is advisable in osteoporotic bone [16]. Expandable [21] or barbed [34] designs are important improvements compared to simpler implants, and our implant follows this path. In addition, the double-external thread and the two-barrette leg improve our implant’s primary stability. Like the implant used by Bencharit et al. [7], our implant’s middle implant body porosity is designed to improve osseointegration. The primary advantage of titanium is reduced manufacturing costs [35].

Our dental implant’s double 1 mm thread external threaded area increases the space between the thread pitches, removes less bone during implant insertion, reduces the heat generated during insertion, and improves the insertion torque in low-density osteoporotic bone. These features are advantageous [36,37].

Our dental implant–abutment connection area has an angle of 16° because when two conical metal parts fit together with an angle of 8° or less, a wedge effect is produced due to the friction between the two walls [38].

The implant–abutment Morse taper fixation connection area, which we used in our dental implant, reduces the micro gap between the implant and abutment connection since it increases the contact surface between them. This feature minimizes bacterial microleakage [39], decreases bone tissue reabsorption [40], and stabilizes soft tissues [41] with a better long-term aesthetic result [42]. In addition, the Morse cone connection absorbs the vibration and pressure exerted on the abutment, avoiding loosening the prosthetic screw [43].

The major issue faced by any dental implant is resistance to daily chewing forces. The first issue is the force any dental implant must stand daily. Scientific reviews have analyzed the maximum chewing forces and compared these with the static tests, reporting a vast variation depending on the age, sex, measurement system, and measurement method used [44,45,46]. In addition, as people age, osteoporosis is commonplace, markedly influencing primary and secondary implant stability with a concomitant loss of resistance to chewing force [47,48].

The mean reported values for the maximum occlusal force in the intercuspal position are 511.7 N for men and 442.4 N for women [49]. With aging, these forces decrease somewhat (391 N and 203 N, respectively) [50]. The maximum bite force in the molars of female patients with osteoporosis was 117 N and 230 N for those without this medical condition [47]. In male patients, these values are near twice that of their female counterparts (≈240 N). This maximum force value of 240 N for patients with osteoporosis is much lower than the elastic limit and failure force obtained in the static test of our dental implant (663.21 ± 54.23 N). Thus, our implant has sufficient properties compared to these forces.

The reported chewing forces found peak values between 5 and 54 N for incisor and canine [51,52,53,54,55] and 50 and 284 N for premolars and molars [53,54,56]. Our dental implant fatigue limit is 132.60 ± 10.4 N, higher than the occlusal force reported for incisor and canine teeth, but between occlusal force values for premolars and molars for healthy patients.

The general estimation is that one million cycles are equivalent to a year of everyday chewing (365 days, three meals a day, chewing 15 min per meal at 1 Hz, and 60 chews per minute), so the testing performed covers an implant life of at least five years [57]. In addition, the standard itself (ISO 148201:2017) determines the maximum load at which failure does not occur with an infinite (in practice, very large) number of cycles. Therefore, the standard assumes everlasting life, provided the load does not exceed the limit. This phenomenon corresponds to other authors’ infinite fatigue life criterion [58].

This study has allowed us to verify that our dental implant manufacture and functionality are possible, meeting the resistance, stability, and elasticity standards expected for any dental implant. However, we need live animal tests and human clinical studies to confirm these data.

## 6. Strengths and Limitations

Our study was performed following international standards and with certified, calibrated equipment. However, our study has some limitations, First, the number of tested dental implants (thirty-five) was limited. Second, testing did not address bruxism, i.e., sideways mandible movements. We attempted to simulate this condition by applying loads at a 16° angle, but this might not entirely reproduce the clinical scenario.

## 7. Conclusions

A new dental implant designed for placement in osteoporotic bone showed adequate results when tested against ISO 14801. Specifically, the elastic assembly limit resulting from the static test was 463.94 Nm, and the mean maximum failure force value is 663.21 N.

The implant, abutment, and screw set fatigue limit was 132.60 N. These results are satisfactory for incisor and canine teeth but between the values for premolars and molars for healthy patients.

The system subjected to the 132.60 N load level exceeded the five million cycles stipulated by the standard. Therefore, the assembly ensures infinite life against loads below the fatigue limit.

## Figures and Tables

**Figure 1 materials-16-02228-f001:**
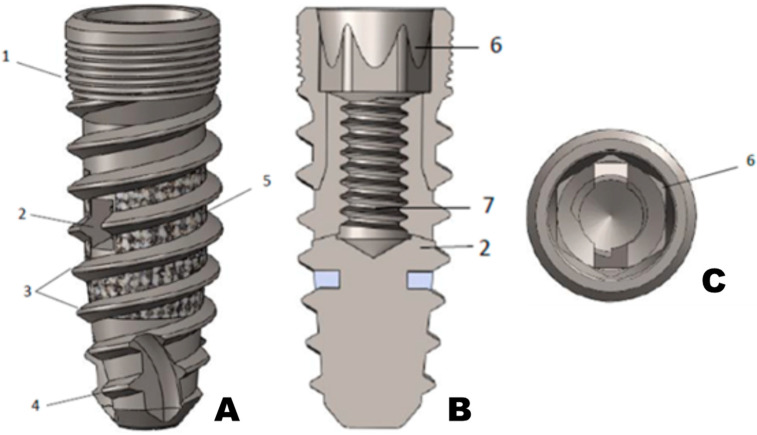
Dental implant body. (**A**) outside screw appearance, (**B**) coronal section, and (**C**) top view.

**Figure 2 materials-16-02228-f002:**
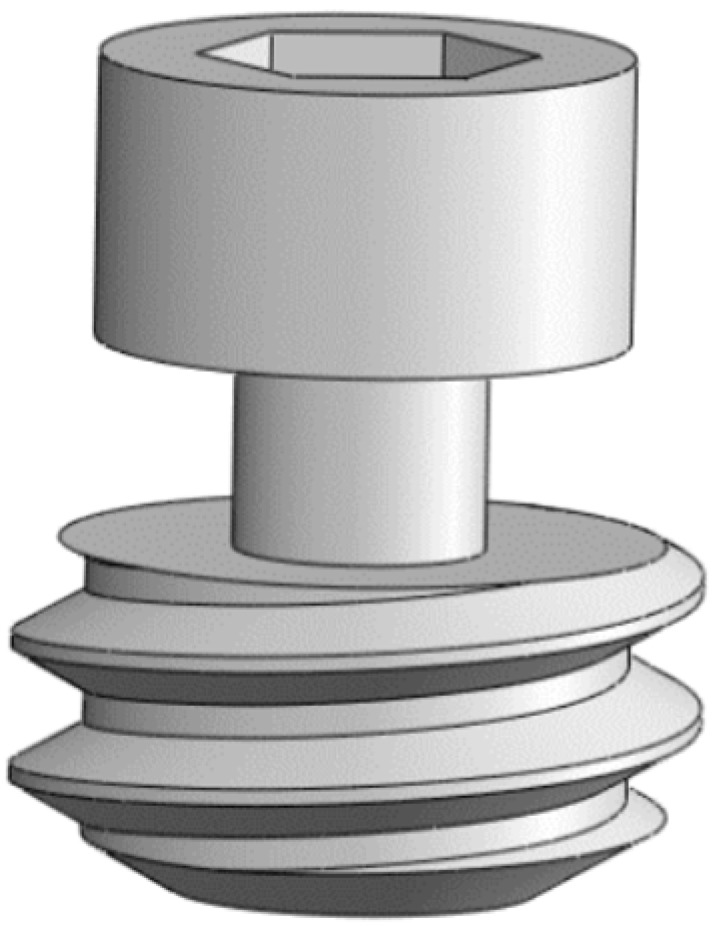
Deployment screw.

**Figure 3 materials-16-02228-f003:**
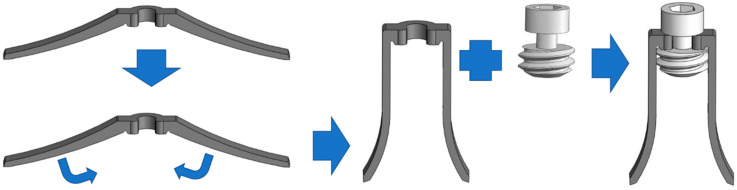
Barrette and insertion screw.

**Figure 4 materials-16-02228-f004:**
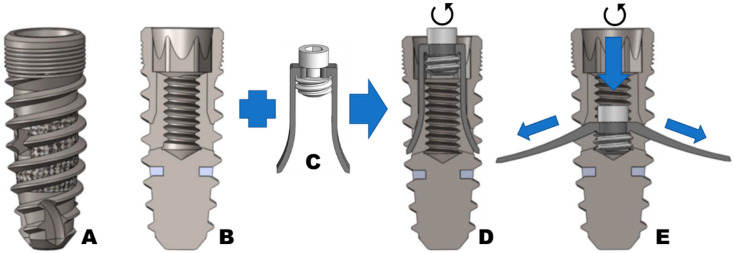
(**A**) Outside dental implant body appearance, (**B**) coronal section, (**C**) barrette and deployment screw, (**D**) barrette and deployment screw inserted inside main body dental implant, and (**E**) barrette deployment with externally deployed legs.

**Figure 5 materials-16-02228-f005:**
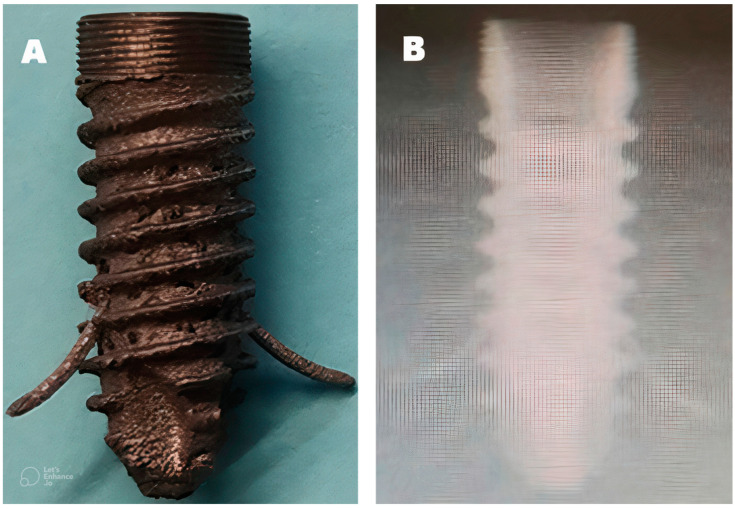
(**A**) Dental implant exterior with the barrette deployed. (**B**) X-ray image once inserted inside a cadaveric human osteoporotic jaw.

**Figure 6 materials-16-02228-f006:**
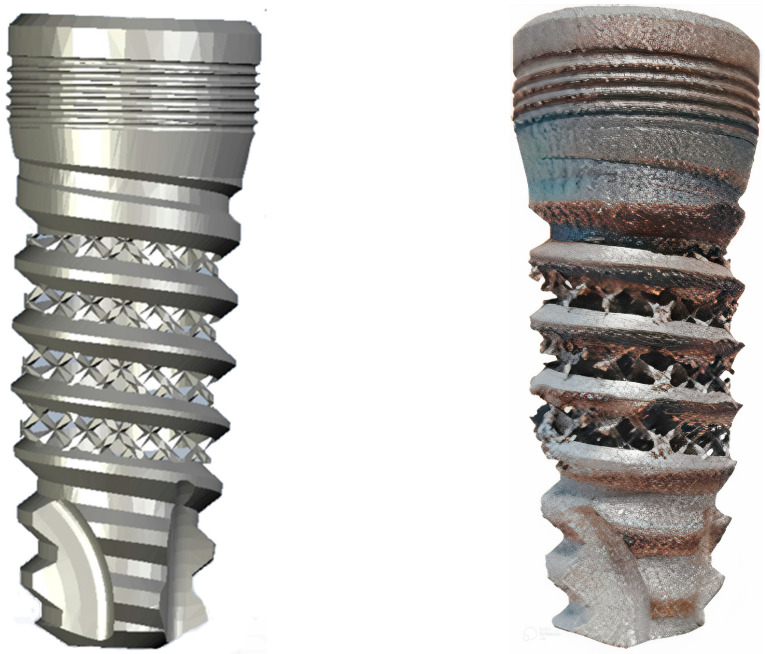
Implant body manufactured by additive 3D printing before machining its internal thread. Left side image, computer recreation. Right side the 3D printed implant.

**Figure 7 materials-16-02228-f007:**
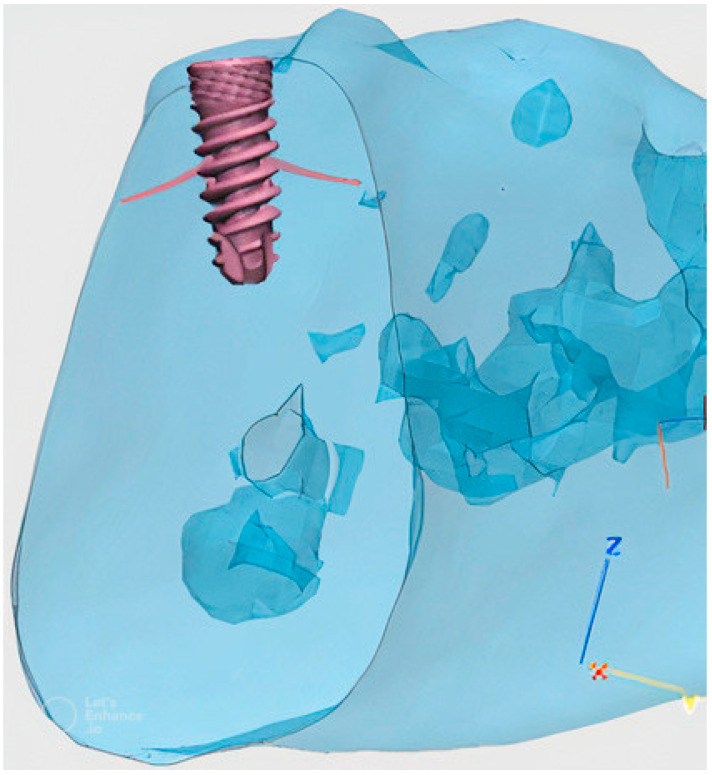
Artistic rendition of our dental implant inserted in a human jaw section reconstruction.

**Figure 8 materials-16-02228-f008:**
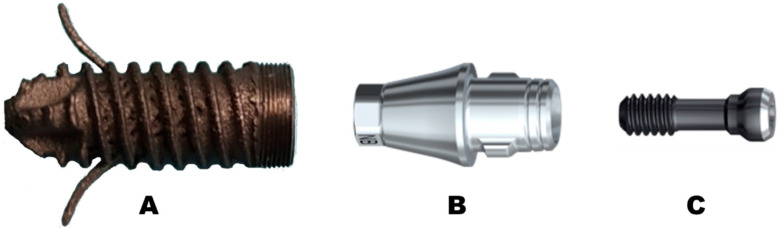
(**A**) The dental implant, (**B**) universal conical base connection, and (**C**) securing screw.

**Figure 9 materials-16-02228-f009:**
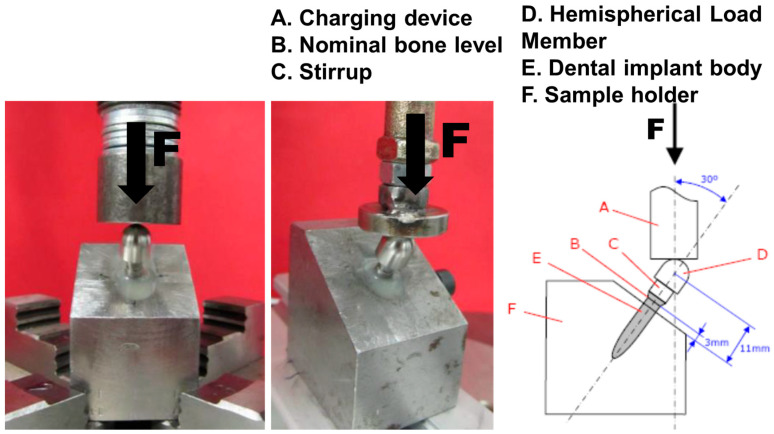
Dental implant in the experimental setting.

**Figure 10 materials-16-02228-f010:**
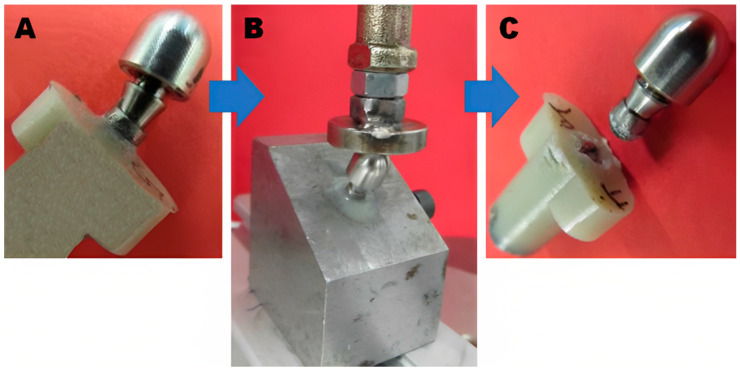
(**A**) Initial status with the dental implant in place and connected with the actuator, (**B**) the system mounted in the testing machine, and (**C**) the system after the test with its failure.

**Figure 11 materials-16-02228-f011:**
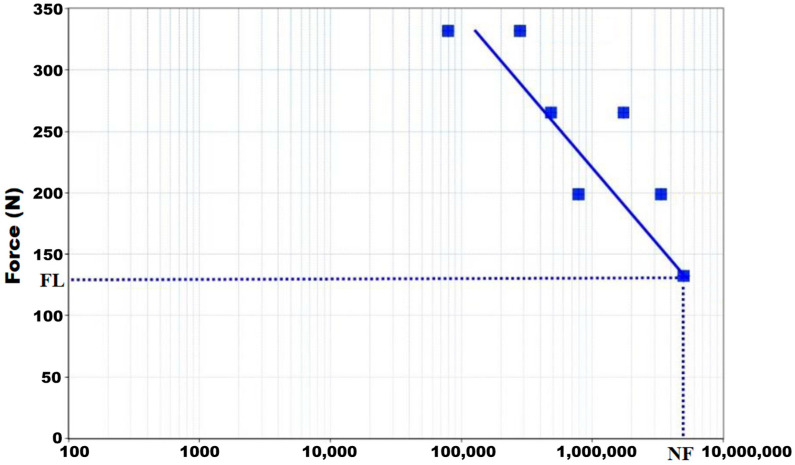
Load cycle diagram. LF: Fatigue limit. NF: Number of cycles without failure.

**Table 1 materials-16-02228-t001:** Maximum loads that we applied with its amplitude and preload.

Maximum Applied Load (N)	Amplitude (N)	Preload (N)
331.60	298.40	33.20
265.30	238.80	26.50
199.00	179.10	19.90
132.60	119.30	13.30

**Table 2 materials-16-02228-t002:** The number of cycles supported by our dental implants for each load.

Maximum Applied Load (N)	Number of Cycles
417.70	79,423
331.60	280,841
265.30	487,551
223.40	1,731,994
199.00	783,023
167.50	3,351,847
132.60	5,000,000

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
