# Peer review of "Barbed Dental Ti6Al4V Alloy Screw: Design and Bench Testing"

_materials, 2023, doi:10.3390/ma16062228_

Round 1

Reviewer 1 Report

Through an innovative implant and central screw design, researchers devised a barbed implant that might strengthen implant stability in loose alveolar bone. The most striking feature of the implant designing concept is the anchoring system and the mechanically extended barrette. Also, its neck thread, self-drilling thread, porous surface design are mature and reasonable. The results are acceptable in static and flexion-compression fatigue tests. Overall, the study is novel and the design concept of auxiliary retainer system that does not hinder implantation procedure or implant strength is refreshing.

Nevertheless, some questions remain to be answered or corrected in the paper:

1.      The most important question for this implant is: does the anchoring system enhance primary implant stability in osteoporotic bone? It awaits yet to be proved by in vitro experiments or 3D finite element analysis.

2.      Can the barrette successfully extend through osteoporotic bone? In vitro experiments or 3D finite element analysis are recommended.

3.      Given the delicate size of the barrette, its flexure strength needs to be evaluated separately to demonstrate that it will not fracture during extension. Or answer the required minimum flexure strength in theory?

Author Response

To Reviewer 1:

Thank you very much for revising our manuscript. Your comments are fundamental to improving its quality. Therefore, we reply below in red color.

Professor V. Vanaclocha

 (x) English language and style are fine/minor spell check Comments and Suggestions for Authors

We have revised the manuscript again, but we are aware that further professional assistance will be needed before publication.

Through an innovative implant and central screw design, researchers devised a barbed implant that might strengthen implant stability in loose alveolar bone. The most striking feature of the implant designing concept is the anchoring system and the mechanically extended barrette. Also, its neck thread, self-drilling thread, porous surface design are mature and reasonable. The results are acceptable in static and flexion-compression fatigue tests. Overall, the study is novel and the design concept of auxiliary retainer system that does not hinder implantation procedure or implant strength is refreshing.

Nevertheless, some questions remain to be answered or corrected in the paper:

  1. The most important question for this implant is: does the anchoring system enhance primary implant stability in osteoporotic bone? It awaits yet to be proved by in vitro experiments or 3D finite element analysis.

We designed the implant specifically for osteoporotic bone. We have done a study on the calf rib bone, and we are currently analyzing the results. The idea is to repeat the study in human cold-preserved mandibular osteoporotic edentulous cadaveric specimens

  1. Can the barrette successfully extend through osteoporotic bone? In vitro experiments or 3D finite element analysis are recommended.

In the calf rib bone, the barrette can extend through the cancellous bone without problem in any tested specimens. Data are pending final analysis before submitting them for publication.

  1. Given the delicatesize of the barrette, its flexure strength needs to be evaluated separately to demonstrate that it will not fracture during extension. Or answer the required minimum flexure strength in theory?

We have not done these tests yet.

Reviewer 2 Report

This manuscript designed a new type of implant with barbed dental Ti6Al4V alloy screw. The original idea is interesting. However, the feasibility of this implant was questionable. With the barrette being inserted into the implant, is it possible to be pushed towards the implant's exterior? The alveolar bone is hard tissue and will not allow easy penetration without proper osteotomy.

Without the barrette design, the implant is not quite different from the existing implant designs. Another concern is with the porous design and holes where the barrette will come out, the strength of the implant will be compromised. 

The writing of the manuscript needs to be improved, academic style of writing would be appreciated. 

Author Response

To Reviewer 2:

Thank you very much for revising our manuscript. Your comments are essential to improving its quality. Therefore, we reply below in red color.

Professor V. Vanaclocha

 (x) Moderate English changes required
We have revised the manuscript again, but we are aware that further professional assistance will be needed before publication.

This manuscript designed a new type of implant with barbed dental Ti6Al4V alloy screw. The original idea is interesting. However, the feasibility of this implant was questionable. With the barrette being inserted into the implant, is it possible to be pushed towards the implant's exterior? The alveolar bone is hard tissue and will not allow easy penetration without proper osteotomy.

The idea is that the barrette rests on the nearby cortical bone after going through the cancellous part. We do not intend to penetrate the cortical alveolar bone. We now provide an x-ray that shows it. We have tested the implant, specifically the barrette’s deployment in the calf rib bone, with no problem whatsoever. These data are pending analysis, and we intend to submit them in another manuscript.

Without the barrette design, the implant is not quite different from the existing implant designs. Another concern is with the porous design and holes where the barrette will come out, the strength of the implant will be compromised. 

We have done flexion-compression fatigue tests that show the implant has enough resistance as a dental implant. The failures happened at the actuator-implant interface and not at the implant itself, as we report in the manuscript.

The writing of the manuscript needs to be improved, academic style of writing would be appreciated.

Professional English editing is costly. However, we are ready to pay for it once the manuscript is accepted for publication.

Reviewer 3 Report

Reviewer Comments

Manuscript ID: materials-2205621

Title: Barbed Dental Ti6Al4V Alloy Screw: Design and Bench Testing  
Authors: Keila Lovera-Prado, Vicente Vanaclocha *, Carlos M Atienza, Amparo
Vanaclocha, Pablo Jorda-Gomez, Nieves Saiz-Sapena, Leyre Vanaclocha
Submitted to section: Biomaterials,
Special Issue: Dental Materials and Devices: Volume II

The paper describe a new dental implant with improved primary stability features.

The paper is rather a mechanical approach of the implant  material than an accurate characterization of material itself, which must take int account not only the mechanical shape but the real interactions between material and body fluids surrounding the implant.

By point of view of mechanical approach the applied methods are adequately described.

By point of view of materials / environment interface there are missing experiments and methods.

It could be a better view of implant stability if the authors give some results of testing this materials shape of implant at the interface with surrounding environment by point of view of chemical-electrochemical stability, not only mechanical one.

Conclusion:

The designed dental implant prototype is enough described but point of view of mechanical approach.

The paper could be accepted for publication after minor revision (corrections to minor methodological errors and text editing) decided by the Editor.

Author Response

To Reviewer 3:

Thank you very much for revising our manuscript. Your comments are an immense addition to improving its quality. Therefore, we reply below in red color.

Professor V. Vanaclocha

The paper describe a new dental implant with improved primary stability features.

The paper is rather a mechanical approach of the implant material than an accurate characterization of material itself, which must take into account not only the mechanical shape but the real interactions between material and body fluids surrounding the implant.

By point of view of mechanical approach the applied methods are adequately described.

By point of view of materials / environment interface there are missing experiments and methods.

It could be a better view of implant stability if the authors give some results of testing this materials shape of implant at the interface with surrounding environment by point of view of chemical-electrochemical stability, not only mechanical one.

The material that we used for the implant is a Ti6Al4V alloy. This material has known chemical-electrochemical stability when implanted in bone (the mandible or other bones). So that is why we did not repeat any studies in this arena. However, we have conducted a lab study on a cold-preserved calf rib bone and are currently analyzing the data.

Conclusion:

The designed dental implant prototype is enough described but point of view of mechanical approach.

Reviewer 4 Report

The study describes a newly designed implant type and its first, pre-cilnical testing. Although the topic is interesting and raises some important questions, in its current form, it is not even possible to review it properly.

The introduction part is way too short, and does not follow the usual flow of introductions for scientific research articles. Please describe in much more details, how this new type was developed, reasons and references are needed!

In the materials part, there are a lot of unclear details, which needs detailed clarification. Please explain, if the barrette and deployment screw is in the implant, can the cover screw or abutment be inserted, and if yes, how? Detailed description and also real photographs are needed to understand this. Please also add a picture showing the implant with the barrette inside in a real photograph.

Also please add detailed description about the experiment specimen numbers. How many implants have been tested, etc. This need to be described in the materials section.

Author Response

To Reviewer 4:

Thank you very much for revising our manuscript. Your comments are critical to improving its quality. Therefore, we reply below in red color.

Professor V. Vanaclocha

 (x) Moderate English changes required

We have revised the manuscript again, but we are aware that further professional assistance will be needed before publication.

Comments and Suggestions for Authors

The study describes a newly designed implant type and its first, pre-clinical testing. Although the topic is interesting and raises some important questions, in its current form, it is not even possible to review it properly.

The introduction part is way too short, and does not follow the usual flow of introductions for scientific research articles. Please describe in much more details, how this new type was developed, reasons and references are needed!

We have revised and expanded the introduction, adding new references

In the materials part, there are a lot of unclear details, which needs detailed clarification. Please explain, if the barrette and deployment screw is in the implant, can the cover screw or abutment be inserted, and if yes, how? Detailed description and also real photographs are needed to understand this. Please also add a picture showing the implant with the barrette inside in a real photograph.

We can insert the abutment or cover screw once the deployment screw is fully screwed and the barrette’s fins are already outside the implant’s body.

We have added new figures.

Also please add detailed description about the experiment specimen numbers. How many implants have been tested, etc. This need to be described in the materials section.

We tested thirty-five dental implants.

Reviewer 5 Report

The authors are requested to:

1- Elaborate the exact components of the titanium used for manufacturing the implant.
2- Elaborate if this design is patented or not.

3- Describe the steps for manufacturing the implant.

4- figure 6 is out of focus and should be replaced by high quality images.

5- The introduction should be expanded to clarify the points of innovation in the authors' implant.

6- The actual dimensions of the barrette is not mentioned.  If figure 5 is comparable to the actual dimensions so this system needs a bulk of bone on the buccal and lingual sides.

7- The authors should clarify the mechanism of penetration of the barrette in the bone (i.e. is it by a sharp tip penetration.).  Is there any risk of bone perforation in case of thin alveolar bone ridge.

Author Response

To Reviewer 5:

Thank you very much for revising our manuscript. Your comments are invaluable to improving its quality. Therefore, we reply below in red color.

Professor V. Vanaclocha

Yes

Can be improved

Must be improved

Not applicable

Comments and Suggestions for Authors

The authors are requested to:

1- Elaborate the exact components of the titanium used for manufacturing the implant.

Ti6Al4V alloy. Now it is stated in the manuscript.

2- Elaborate if this design is patented or not.

We patented our dental implant in Europe, the USA (Patent number 16/956594), and Brazil. We have included this information in the manuscript

3-Describe the steps for manufacturing the implant.

We use additive 3D printing to manufacture the barrette and the dental implant's body to achieve the middle zone porosity of the latter. Next, we machine the implant's body internal thread as well as the deploying screw and give it an Allen hexagonal head

4- figure 6 is out of focus and should be replaced by high quality images.

We have improved its quality.

5- The introduction should be expanded to clarify the points of innovation in the authors' implant.

We have expanded it.

6- The actual dimensions of the barrette is not mentioned.  If figure 5 is comparable to the actual dimensions so this system needs a bulk of bone on the buccal and lingual sides.

This barrette is 1.5mm in diameter, has a blunt tip on both sides, and comes out 4mm at each side from the implant once deployed. Figure 5 (now figure 7) is an artist's rendition that is not accurate. The normal mandible accommodates our dental implant with its barrette with no problem.

7- The authors should clarify the mechanism of penetration of the barrette in the bone (i.e., is it by a sharp tip penetration.).  Is there any risk of bone perforation in case of thin alveolar bone ridge.

The barrette's tip is blunt and penetrates only the cancellous bone. Therefore, the is no risk of cortical alveolar bone penetration.

Round 2

Reviewer 1 Report

Reviewer Comments:

This study attempts to address the lack of initial stability after implantation in osteoporosis patients through a new implant design. Several reasonable designs were utilized such as a porous implant surface, morse taper connection, neck thread, self-tapping tip, and a double thread screw. The most interesting of which was the anchoring system: the mechanically extended barrette. This study is well innovative and constructs mechanical models as well as human bones to provide a preliminary answer to the stress limits of this new implant under load, the way it works during implantation, and the pattern of the barrette extending. Thus this study is generally credible and innovative.

Author Response

To Reviewer 1:

Thank you very much for revising our manuscript. Your comments are fundamental to improving its quality. Therefore, we reply below in red color.

Professor V. Vanaclocha

 (x) English language and style are fine/minor spell check Comments and Suggestions for Authors

A professional English native has reviewed and corrected the English language and edition.

Through an innovative implant and central screw design, researchers devised a barbed implant that might strengthen implant stability in loose alveolar bone. The most striking feature of the implant designing concept is the anchoring system and the mechanically extended barrette. Also, its neck thread, self-drilling thread, porous surface design are mature and reasonable. The results are acceptable in static and flexion-compression fatigue tests. Overall, the study is novel and the design concept of auxiliary retainer system that does not hinder implantation procedure or implant strength is refreshing.

Nevertheless, some questions remain to be answered or corrected in the paper:

  1. The most important question for this implant is: does the anchoring system enhance primary implant stability in osteoporotic bone? It awaits yet to be proved by in vitro experiments or 3D finite element analysis.

We designed the implant specifically for osteoporotic bone. We have done a study on the calf rib bone, and we are currently analyzing the results. The idea is to repeat the study in human cold-preserved mandibular osteoporotic edentulous cadaveric specimens

  1. Can the barrette successfully extend through osteoporotic bone? In vitro experiments or 3D finite element analysis are recommended.

In the calf rib bone, the barrette can extend through the cancellous bone without problem in any tested specimens. Data are pending final analysis before submitting them for publication.

  1. Given the delicatesize of the barrette, its flexure strength needs to be evaluated separately to demonstrate that it will not fracture during extension. Or answer the required minimum flexure strength in theory?

We have not done these tests yet.

Reviewer 2 Report

no further comments

Author Response

To Reviewer 2:

Thank you very much for revising our manuscript. Your comments are essential to improving its quality. Therefore, we reply below in red color.

Professor V. Vanaclocha

 (x) Moderate English changes required
A professional English native has reviewed and corrected the English language and edition.

This manuscript designed a new type of implant with barbed dental Ti6Al4V alloy screw. The original idea is interesting. However, the feasibility of this implant was questionable. With the barrette being inserted into the implant, is it possible to be pushed towards the implant's exterior? The alveolar bone is hard tissue and will not allow easy penetration without proper osteotomy.

The idea is that the barrette rests on the nearby cortical bone after going through the cancellous part. We do not intend to penetrate the cortical alveolar bone. We now provide an x-ray that shows it. We have tested the implant, specifically the barrette’s deployment in the calf rib bone, with no problem whatsoever. These data are pending analysis, and we intend to submit them in another manuscript.

Without the barrette design, the implant is not quite different from the existing implant designs. Another concern is with the porous design and holes where the barrette will come out, the strength of the implant will be compromised. 

We have done flexion-compression fatigue tests that show the implant has enough resistance as a dental implant. The failures happened at the actuator-implant interface and not at the implant itself, as we report in the manuscript.

The writing of the manuscript needs to be improved, academic style of writing would be appreciated.

A professional English native has reviewed and corrected the English language and edition.

Reviewer 5 Report

The modifications improved the quality of manuscript 

Author Response

To Reviewer 5:

Thank you very much for revising our manuscript. Your comments are invaluable to improving its quality. Therefore, we reply below in red color.

Professor V. Vanaclocha

A professional English native has reviewed and corrected the English language and edition.

Can be improved

Must be improved

Not applicable

Comments and Suggestions for Authors

The authors are requested to:

1- Elaborate the exact components of the titanium used for manufacturing the implant.

Ti6Al4V alloy. Now it is stated in the manuscript.

2- Elaborate if this design is patented or not.

We patented our dental implant in Europe, the USA (Patent number 16/956594), and Brazil. We have included this information in the manuscript

3-Describe the steps for manufacturing the implant.

We use additive 3D printing to manufacture the barrette and the dental implant's body to achieve the middle zone porosity of the latter. Next, we machine the implant's body internal thread as well as the deploying screw and give it an Allen hexagonal head

4- figure 6 is out of focus and should be replaced by high quality images.

We have improved its quality.

5- The introduction should be expanded to clarify the points of innovation in the authors' implant.

We have expanded it.

6- The actual dimensions of the barrette is not mentioned.  If figure 5 is comparable to the actual dimensions so this system needs a bulk of bone on the buccal and lingual sides.

This barrette is 1.5mm in diameter, has a blunt tip on both sides, and comes out 4mm at each side from the implant once deployed. Figure 5 (now figure 7) is an artist's rendition that is not accurate. The normal mandible accommodates our dental implant with its barrette with no problem.

7- The authors should clarify the mechanism of penetration of the barrette in the bone (i.e., is it by a sharp tip penetration.).  Is there any risk of bone perforation in case of thin alveolar bone ridge.

The barrette's tip is blunt and penetrates only the cancellous bone. Therefore, the is no risk of cortical alveolar bone penetration.
